# Drug delivery process simulation—Quantifying the conformation dynamics of paclitaxel and cremophor EL

Mafiz Uddin[1]*, Dennis Coombe[2]

1 Alberta Computational Biochemistry Lab, Edmonton, Alberta, Canada, 2 Computer Modelling Group, Calgary, Alberta, Canada

☉ These authors contributed equally to this work.
* mafiz.uddin36@gmail.com

## Abstract

Paclitaxel is a highly successful anti-neoplastic cancer drug. The first clinically successful paclitaxl-delivery method is a mixture with cremophor EL and ethanol, here termed the taxol micelle. Until now, molecular dynamics analysis has not been presented to quantify the structural and conformation properties of these drug molecules when interacting with each other to create this nonstandard micelle. Here we apply systematic molecular simulation and statistical analysis of paclitaxel and cremophor EL conformations based on all atom and coarse-grained approaches. The cremophor EL in the taxol micelle showed a clustering network in a 3D landscape where paclitaxel can be loaded at much higher than standard concentration with no aggregation. Paclitaxel particles within the cremophor EL cavities showed some oscillatory behaviour due to a repeated adsorption/desorption with the surrounding network. Paclitaxel conformations at the lowest energy state can be described when the side-chain phenyl rings are closer relative to the immobile core. Cremophor EL molecules reached the highest energy state when wings were fully spread and at the lowest energy state when wings were fully closed. The spiral shapes were observed to be the dominant species in the cremophor EL population. We then established reliable statistical correlations between molecular conformations and the energy states. Our reliable all atom and coarse-grained modelling approach can also be applied for effective drug design analysis using different drug delivery systems.

## Introduction

Paclitaxel (PTX) was approved as a novel anti-neoplastic cancer drug in the early 1990's [1]. Since that time, its use in treating breast, ovarian, lung and other cancers has remained popular. Its mode of action is to stabilize microtubule polymers, blocking cell replication [2]. A major drawback of this drug however is its very low water solubility (0.1 ug/cm³ (1e-7 massfr) or 10 uM (1.8e-7 molefr)) due to its hydrophobic

**Data availability statement:** All relevant data are within the manuscript and its Supporting Information files. The following data set were included (as supporting materials) to replicate all simulation results and study findings reported in the article: S3 File. PTX Data (S3_File.pdf) S4 File. CrEL Data (S4_File.pdf)

**Funding:** The author(s) received no specific funding for this work.

**Competing interests:** The authors have declared that no competing interests exist.

nature as characterized by its octanol-water partition coefficient (log(Kow) = 3.16 (range 2.5–4.4)) which limits its direct applicability [3,4]. Letchford et al. [5] quote an even lower aqueous solubility, (1.17 uM (2.1e-8 molefr)).

A major area of research is the development of appropriate PTX drug-delivery vehicles to increase its effective blood solubility. Here, a wide variety of proposed nano-particle formulations are under investigation, which includes micelles, microemulsions, and block copolymers and liposomes. Various review articles [6] can be accessed to trace the developments and comparative features of such ever-evolving formulations for PTX. These include Abraxane [7–10] and Genexol-PM [11,12] which are currently in clinical use. In particular, Genexol, a 24 nm nanoparticle, exhibits a typical surfactant-induced spherical micelle structure with the polar portion providing an outer corona while the inner nonpolar core is formed from the nonpolar surfactant plus solubilized PTX drug.

Taxol was the first delivery vehicle for paclitaxel which was developed to combat the low solubility of PTX in blood. In fact, Taxol is so widely used that often the terms Taxol and paclitaxel are used interchangeably in the technical literature. In this article we will reserve the name "Paclitaxel" for the cancer active chemical and use the term "Taxol" or "Taxol micelle" to describe its delivery formulation with Cremophor EL (CrEL) and ethanol (EOH) and investigate aspects of its structure. Taxol represents the first clinically successful PTX-delivery method [13,14] and any newly-developed alternate formulation is first compared with the Taxol formulation for efficiency and efficacy [15]. This nanoparticle is a surfactant micelle consisting of a 50/50 mixture of a natural surfactant CrEL (poly-oxyethylene-glycerol tricinoleate 35) and an alcohol (ethanol) which solubilizes an amount of drug [14]. From Gelderblom et al. [16] this mixture is 2.5 ml CrEL, 2.5 ml Ethanol, and 0.03 gm PTX (or 0.5 ml CrEL, 0.5 ml Ethanol, and 0.006 gm PTX). Based on component molecular weights and mass densities at standard conditions, this converts to molar compositions of (7e-6, 8478e-6, 212e-6) moles, for PTX, Ethanol, and CrEL respectively. Note this is roughly equivalent to 7 molecules PTX, 8478 molecules Ethanol, 213 molecules CrEL. We will approximate this as 8 molecules PTX, 7888 molecules Ethanol, 200 molecules CrEL to conveniently build the composition of our standard micelle Taxol system as dicussed in section (methodology).

The pharmacokinetics of Taxol delivery have been extensively studied at the whole-body scale [17–20]. In particular, the role of CrEL on drug toxicity and drug delivery has been extensively emphasized [16]. After careful investigation, this latter effect is attributed to paclitaxel incorporation into surfactant micelles [21–23]. Ultimately, pharmacokinetic models utilizing explicit micelle binding terms are preferred as more mechanistically correct [24–28]. Here we describe the actual structure of Taxol micelles (PTX, EOH, CrEL) and the interaction of PTX and CrEL molecules inside the micelle using MD giving a more complete microscopic understanding of this drug vehicle.

Modern molecular scale investigations (molecular dynamics (MD) - all atom (AA) or coarse grained (CG); dissipative particle dynamics (DPD)) of new drug-delivery methods are now often employed to further study mechanisms. These

include studies of formulations with paclitaxel. Such models are based on protein data bank structures of paclitaxel [29–31]. Its basic anti-cancer mechanism (binding to tubulin) has been studied with MD [32–34]. All-atom simulations [35] of the interaction of paclitaxel (and other drugs) with poly(ethylene glycol) polymer carriers illustrate the mechanistic insight provided by such computational methods. Lim et al. [36] use all-atom methods to study the properties of paclitaxel-chemical adducts. Coarse-grain simulations of a paclitaxel-block copolymer self-assembly [37] and a paclitaxel-conjugate system [38,39] demonstrate aggregation dynamics and concentration effects of paclitaxel in solution. Most particularly, the upscaling methods of Peng et al. [40] and Jiang et al. [41], with a detailed comparison of AA to CG representation of paclitaxel, most closely parallels the computational approach we will apply, as discussed below. Further upscaling provided by dissipative particle dynamics (DPD) allows a more comprehensive analysis of self-assembly of paclitaxel with a block copolymer system [42].

Surprisingly, however, there doesn't appear to have been an investigation of CrEL and Taxol via such computational methods. This is probably because the early clinical development of Taxol predates the rise of the popularity of using these effective numerical methods. Since Taxol remains the accepted standard for paclitaxel drug delivery, we feel that such a study can be useful and informative. Here, we present an approach to the numerical MD modeling of the Taxol formulation at the all-atom and coarse-grained scales, probing mechanisms at various scales. The methods developed here for Taxol modelling are equally applicable to other polymeric micelle carriers for paclitaxel, and indeed to other NP drug system modelling.

A detailed comparison of all-atom (AA) and coarse-graining (CG) modelling approaches to molecular dynamics (MD) simulation of Taxol is presented, based on a comparison up to 50ns. We establish that our PTX models (AA and CG) are consistent with previous literature models. Conversely, the building or synthesizing CrEL liquid models using MD was a major task, which hasn't been investigated previously. We extend this analysis by exploring structure-determining mechanisms in detail utilizing CG methods. The following two standard concentration SC-Taxol micelles systems are explored, without and with a surrounding water bath: (i) 8PTX-7888EOH-200CrEL, (ii) 8PTX-7888EOH-200CrEL-104220W. Additionally, these same two systems at higher paclitaxel loading (high concentration HC-Taxol) are also investigated. These cases are: (3) 32PTX-7888EOH-200CrEL (4) 32PTX-7888EOH-200CrEL-104220W, without and with a surrounding water bath. Details of MD methods and model set-up are presented in section (methodology). The CG simulations for these four cases at longer times (up to 1000ns) are analyzed dynamically. The AA simulations for the two Taxol micelles (without water) at shorter times (up to 50ns) are also analyzed. Both results are presented in section (CG simulation for taxol micelle). To further quantify the underlying micelle structure, we use MD methods (AA and CG) to characterize PTX and CrEL conformational states in section (PTX and CrEL conformations dynamics). These comparisons are evaluated at 50ns for AA and 1000ns for CG systems. Here the physical conformations of PTX and CrEL molecules in the HC Taxol micelles and selected binary systems are analyzed, and provide a full statistical picture of their dynamical properties. Comparison is made with an earlier study of predicted PTX conformations, while the conformational analysis of CrEL appears to have not been done previously.

## Methodology

One of our main objectives is to define all the known initial conditions in setting the FF topology and the Taxol simulation system. The initial conditions are bonded parameters (such as bonds, angles and dihedrals) and a correct physical state of the Taxol system.

### MD simulation setup

General introduction on the use of molecular dynamics can be found in standard textbooks [43–44]. For our simulations, we use the OPLSA FF (force field) [45] for the AA MD simulations. In CG, particles and the non-bonded interaction parameters were chosen from the MARTINI FF [46] which employs 18 bead types. We set up CG networks for paclitaxel and

Cremophor EL by incorporating the AA structural details (bonds, angles, dihedrals). Our methodology here is divided into three main steps. In step one, CG mapping for the 4 model components (PTX, CrEL, EOH, H2O) along with some basic numerical simulation parameters are presented. The CG networks for PTX and CrEL are set up by using the exact details of AA molecular structures (or parent template). In step two, Taxol micelle simulation cases are initialized with an innovative technique to produce a correct physical state for this highly contrasted ternary mixture (heavier and lighter molecules). This approach can be applied for many other complex heterogeneous systems. In step three, we have applied a methodology to establish a statistical correlation between molecular conformations and conformations energies in the Taxol micelle.

**Setup AA and CG molecular topologies.** The AA molecular coordinates are obtained and then optimized by energy minimization simulation. The CG bead coordinates are calculated by taking the mass weighted average of the heavier atoms coordinates (building block as setup in CG mapping). This preserves the overall density of the AA system. The AA and CG FF typologies and the general properties of these molecules are summarized below:

(i) Paclitaxel (PTX) – The atomic coordinates were obtained from several studies on NMR and X-ray analyses of PTX crystal [29–31]. Fig 1 shows the AA and CG mappings for paclitaxel. This molecule has several polar, apolar and benzene groups. As a result, this has mixed hydrophobic and hydrophilic properties. In this study, we will quantify the difference between aggregation dynamics in water (highly polar) and ethanol (intermediate polarity). We note that the paclitaxel molecule ($C_{47}H_{51}NO_{14}$) is nearly spherical with little to no flexibility. There are therefore no concerns about the molecular flexibility issues in CG mapping. A total of 24 CG beads were selected in order to preserve all the paclitaxel functional groups. (AA and CG coordinates data in S3 File (given as supporting information), comparable to Peng et al. [40] and Jiang et al. [41])

(ii) Cremophor EL (CrEL) – AA CrEL coordinates were defined based on the CrEL building blocks (glycol, PEG, ricinoleic acid) topology data and then optimized by energy minimization. Fig 1 shows the AA and CG mappings for CrEL. The CG mapping uses 13 polar head beads and 6 beads for a relatively short apolar tail. This is a surfactant with significant head to tail imbalance. We may therefore expect an inward head or outward tail micelle formation tendency in both solvents water and ethanol. Furthermore, the actual CrEL molecule ($C_{126}H_{242}O_{45}$) has 12 repeating polyethylene glycol (PEG) units. As a result, it has significant molecular flexibility and probable folding tendency in the aggregation process in a real system. We assume our 19 bead long CG particle has sufficient flexibility in capturing the self assembly of actual CrEL molecule (AA and CG coordinates data in S4 file (given as supporting information).

There are two possible CG mapping options in describing CrEL molecules, differing with respect to the repeat (PEG) units and glycerol connecting beads. An alternate representation to that of Fig 1 would replace the P1, P2, P3 beads with an N0 type. There has been some discussion of CG representation of PEG in the literature using both representations [47,48]. Here, we evaluated both possible CG mappings via short time comparative simulations without any noticeable variation in results (not shown).

(iii) Solvent Ethanol (EOH) – In CG MARTINI, we chose a single bead intermediate polar particle (P2 type). MD simulations for all the cases presented here used this single polar particle. The EOH molecule ($CH_3CH_2$-$OH$) has both apolar and polar properties. As a further investigation, we also setup 2 beads for ethanol: one for apolar ($CH_3CH_2$) and one for polar ($OH$), (not shown explicitly)

(iv) Water – In AA simulation, we have chosen a simple SPC water model. The CG MARTINI FF proposed a single bead water model representing four AA water molecules along with antifreeze water particles. The CG MARTINI water model was adopted for our model cases with 10% antifreeze water particles.

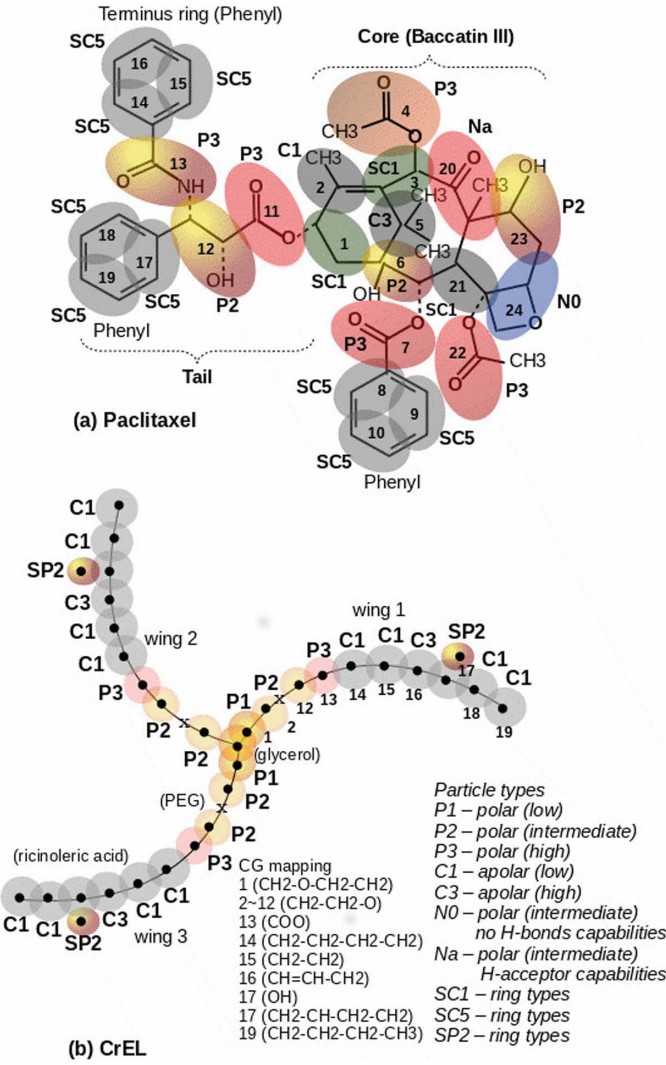

**Fig 1. CG mapping showing atomic groups and particles types, (a) paclitaxel network over AA structure (24 beads), (b) CrEL network (19 beads per wing).**

### Non-bonded and bonded parameter choices.

(i) Non-bonded AA and CG choices: Based on published data for similar types of atoms and charge group (OPLSA, GROMOS), this study was able to define the Lennard-Jones (LJ) and Coulomb potentials parameters for the CrEL and the PTX molecules. In our MD simulations, we use a geometric average option in constructing the parameter matrix for the non-bonded LJ parameters. The non-bonded potential was modified by a shift function in order to improve the cut-off effects. Fig 1 shows the non-bonded LJ parameters for 12 interacting particle types in PTX-CrEL-EOH-H2O model systems. The CG interaction parameters were taken from the published MARTINI FF.

(ii) Bonded parameter choices: The bond stretching between two covalently bonded atoms in AA or beads in CG simulations was represented by a harmonic potential as $V_b = \frac{1}{2}k_b(r-b)^2$. The bond-angle variations between a triplet of atoms in AA and beads in CG were represented by G96 cosine-based angle potential, $V_a = \frac{1}{2}k_\theta\left(cos\theta - cos\theta_0\right)^2$,

for AA and a harmonic potential, $V_a = \frac{1}{2}k_\theta \left(\theta - \theta_0\right)^2$ for CG, respectively. For dihedrals, we applied proper dihedrals using the Ryckaert-Bellemans function in AA, $V_{rb} = \sum_{n=0}^{5} C_n \left(\cos\left(\varphi - 180^0\right)\right)^n$, and standard periodic proper dihedrals in CG, $V_d = k_\varphi \left(1 + \cos\left(n\varphi - \varphi_0\right)\right)$, where, $\varphi$ is the angle between the two planes, with zero corresponding to the *cis* configuration. It should be noted that no restrictions such as improper dihedrals were considered in this study.

**MD numerical parameters.** GROMACS 5.0.7 simulation package [49] was employed for all molecular dynamics simulations. Resulting spatial visualizations use the VMD package [50]. Each simulation system was initialized with an energy minimization step to reduce excessive forces on any one atom. This was accomplished using a steepest descent algorithm, with convergence achieved once the maximum force on any one atom was less than 100 *kJ mol⁻¹nm⁻¹*. After this initialization, the NPT simulations were performed using the Berendsen barostat and Berendsen thermostat with coupling time constants of 1.0 and 0.1 *ps*, respectively. The simulations were also checked with v-rescale for pressure and c-rescale for temperature couplings. The bond lengths of the surfactants were constrained by the LINCS algorithm and those of the water molecules by SETTLE. A grid-based twin-range cutoff for non-bonded interactions of 1.0/1.0 *nm* was used with a time step of 2 *fs* and with a neighbour list update every 10th step. The full particle-mesh Ewald method (PME) was employed for the long-ranged electrostatic interactions with cubic interpolation order of 4 and grid spacing for fast-Fourier-transform (FFT) of 0.16. The Verlet cut-off scheme was employed using the default Verlet buffer size set to 0.005 *kJ/mol/ps* energy drift per atom. It is assumed that the total energy drift occurring in the system is usually much smaller than this default setting for the estimate.

The bonded and non-bonded parameters for all residues produce physically accurate representations in the model systems. Lorentz-Berthelot mixing rules for van der Waals interactions were utilized in simulating all the interactions. The simulation parameters here are very much standard. There are no numerical effects on the simulation results.

## Taxol micelle simulation setup

The following flow diagram summarized how we set up the Taxol simulations cases by aligning with drug design steps (Fig 2). AA MD simulation for the ternary mixture case (8PTX, 7888EOH, 200CrEL) was carried out for 50ns. CG model was set up using the AA trajectories at 10ns and simulation carried out until it reached equilibrium at 1000ns. In the final follow up case, 104220 CG water particles (equivalent to 416,880 water molecules) were added in a cubical water bath by centre positioning the simulated ternary mixture trajectories at 1000ns and simulation continued another 1000ns. The CG simulation here significantly speeds up the computational time range. Total computational time for 50ns AA MD and 2000ns CG simulations are approximately the same.

(1) AA MD - The main task here was to set up a reliable initial mixing condition (8PTX, 200CrEL, 7888EOH). The initial mixing is vitally important when dealing with a mixture of relatively smaller (H2O, EOH) and much larger (PTX, CrEL) molecules. In MD, there is no high energy convection process. In the laboratory, drug mixing is typically done by a stirring process. Equivalent to such a laboratory protocol, we have applied a mixing strategy which significantly reduced MD simulation time.

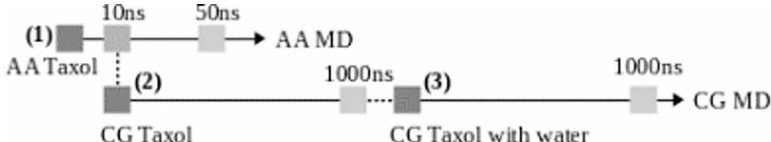

**Fig 2. Flow diagram showing AA and CG simulations strutures.**

We first created 4 identical realizations of a CrEL phase with a correct density of 1.05 gm/cc. For each realization, 50 of AA CrEL molecules with an identical configuration were randomly distributed in each cubical MD simulation box. The NPT simulation was then conducted until it reached an equilibrium density of 1.05 gm/cc.

Secondly, we randomly mixed 2 paclitaxel with 1997 ethanol in a cubical MD simulation box. Again 4 realizations were created. The NPT simulation was performed in order to bring the ethanol system density in equilibrium. In this equilibrium simulation, atomic positions of the paclitaxel molecules were restrained to prevent aggregation.

Finally, a ternary mixture model was set up by mixing 4 boxes of CrEL and 4 boxes of solvent ethanol containing dissolved paclitaxel as shown in Fig 3. Here, the boxes were planted diagonally for maximizing the initial mixing by keeping 0.10 nm inter-phase gap to avoid atoms overlapping. Taxol AA MD simulation then followed an energy minimization step.

(2) CG MD - CG model was then set up using AA simulation trajectories at a selected time of 10ns. The AA coordinates at 10ns were converted to CG initial bead coordinates based on the atomic grouping scheme (as given in Fig 1). The CG bead coordinates (centre of a group of atoms) was calculated as: $r_c^j = \frac{\sum m_i r_i}{\sum m_i}$. Here, $m_i$ and $r_i$ are the atomic mass and coordinates in a given group involved in the summation. For the CG mapping in Fig 1, there are total 24 and 57 atomic groups for PTX and CrEL molecules, respectively. The group index, $i$, for a single bead ethanol ($C_2H_6O$) is 9.

Using the bead coordinates, the vector between any two beads can be expressed in standard orthogonal basis as: $\vec{v_1} = (x_2 - x_1)\,\vec{i} + (y_2 - y_1)\,\vec{j} + (z_2 - z_1)\,\vec{k}$. The bonds (such as vector norm $\|\vec{v_1}\|, \|\vec{v_2}\|$) and angles (such as angle between $\vec{v_1}$ and $\vec{v_2}$, $\theta_{123} = \cos^{-1}(\vec{v_1}.\vec{v_2}, \|\vec{v_1}\|\,\|\vec{v_2}\|)$) for the CG molecular networks were calculated. The dihedral angle, $\xi_{1234}$, between planes (123) and (234) can be defined in vector dot and cross-product notations as: $\xi_{1234} = \frac{180}{\pi} atan2\,(\|\vec{v_2}\|\,\vec{v_1}.\,(\vec{v_2}x\vec{v_3})\,,\,(\vec{v_1}x\vec{v_2})\,.\,(\vec{v_2}x\vec{v_3}))$, here, $\vec{v_1}, \vec{v_2}, \vec{v_3}$ are the vectors between beads (1,2), (2,3) and (3,4), respectively. In the FF topology for PTX and CrEL, we have defined bonds, angles and dihedrals by taking averages over all 8PTX and 200CrEL molecules. Our claim here is that this reduces some uncertainties in CG modelling. This conversion of the AA trajectories is a deterministic method which exactly defines the initial distribution of the molecules, their molecular structures and bonded parameters. The initial mixing condition for the HC Taxol system (32PTX, 200CrEL, 7888EOH) was set up analogously. In our earlier studies [51], this computational approach was first applied to setup CG network for protein molecular structure.

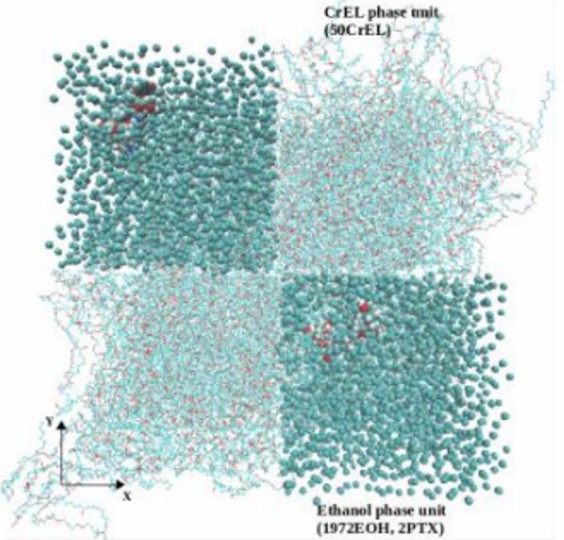

**Fig 3. AA MD setup for micelle of 8 paclitaxel, 200 CrEL and 7988 ethanol after energy minimization (CrEL and paclitaxel containing ethanol phases diagonally inserted)** *(PTX-vdw, EOH-beads, CrEL-lines, O-red, C-cyan).*

## Molecular trajectories and conformation analysis

In order to quantify the MD trajectories at equilibrium within the Taxol micelle, the changes of the following variables were studied over the simulation times (every 2ns for AA and 10ns for CG): component spatial distribution, partial density, energy components, radial distribution function, mean square displacement and radius of gyration.

For our conformation analysis, we have isolated each molecular conformations (PTX, CrEL) from the simulated HC Taxol at 50ns for AA and 1000ns for CG. We then calculated global minimum potential energies for each conformation using the energy minimization algorithm. Afterwards, we were able to develop statistical correlations between conformations and conformations energies. It should be noted that potential energy is not a gauge invariant. Therefore we defined relative energies by subtracting reference energy of the simulated systems. We believe that we are the first in providing a systematic statistical analysis for drug component conformation properties. The details of this methodology are given in conformation section (PTX and CrEL conformations dynamics).

## CG simulation for taxol micelle

This section presents the CG Taxol simulations results for the SC Taxol micelle. CG Taxol simulation was conducted 1000ns until it reached equilibrium at 1000ns. CG Taxol with water simulation was then followed for another 1000ns. In order to assess the MD simulation trajectories, the following variables were analyzed at every 2ns for AA and 10ns for CG over the simulation times: component partial density, energy data, radial distribution functions, mean square displacements and radius of gyration. The components' spatial distributions were visually observed by VMD to quantify any effect of particle polarity on the aggregation. This analysis showed that CG Taxol simulation reached a steady state at 1000ns. There are no noticeable changes observed over 700–1000ns. We used the MD simulations trajectories over 800–1000ns for our final analysis. This illustrated the role of various forces and internal motions inside the Taxol micelles.

## SC micelle structure at equilibrium state

Fig 4 shows a snapshot of the SC Taxol atomic coordinate trajectories at equilibrium, also highlighting a local location with 3PTX and 10CrEL molecules. (VMD utilizing periodic boundary conditions extend CrEL wings outside equilibriated cubic simulation box). A thorough analysis of the spatial snapshot at different view points showed that the CrEL self-aggregated into well defined tails clusters inside the micelle. The CrEL tails at the boundaries did not cluster, but instead pointed outward. The analysis demonstrated that the paclitaxel molecules are in dispersed phase (no aggregation) within the CrEL cluster. The CrEL self aggregation is further re-organized in water (as shown in the next section). In particular, we illustrate how the hydrophobic tails at the micelle boundaries are forced to aggregate.

The partial densities of ethanol, CrEL and paclitaxel along the X-axis are shown in Fig 5 (the Y-, and Z-axis distributions are very similar). The plots show no significant changes in the partial densities at times of 800, 900, and 1000ns. Fig 6 showing bead-bead radial distribution functions (RDFs), can be used to describe the CrEL-PTX aggregation process. This indicates that individual PTX molecules are predominately surrounded by CrEL tail groups, as the three tail groups of CrEL wings (Gy1, Gy2, Gy3) have a local RDF maximum around PTX, while the three head groups of CrEL wings (Gy1, Gy2, Gy3) are excluded from this region. This local nonrandom behaviour contrasts with purely random distribution seen at large distances of the RDFs.

The steady state Taxol micelle as discussed above (Fig 4) was next hydrated with 416,800 water molecules (equivalent 104,220 CG water particles) in a cubical box. CG simulation was conducted another 1000ns. Fig 7 shows a snapshot of the atomic coordinate trajectories at 1000ns. This highlighted the paclitaxel distribution and CrEL aggregation pattern. The simulation results show that 7888 ethanol particles uniformly mix with the surrounding water (as expected), CrEL self-aggregated (formed well defined tails clusters), while the 8 PTX remain trapped (mainly dispersed) within CrEL. This varied potential landscape is likely due to the existence of CrEL molecules at different energy states. This furthermore

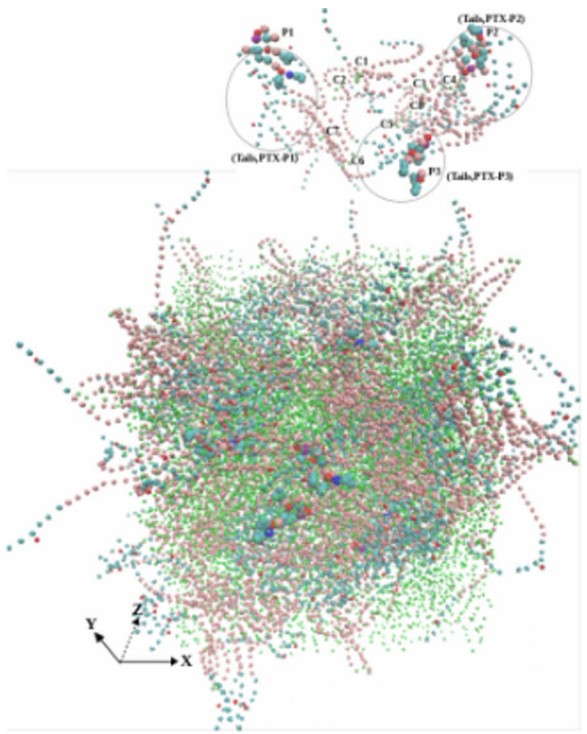

**Fig 4. Taxol micelle at 1000ns (an internal spot above showing 3PTX and 10CrEL) (*CrEL head - red, CrEL tail – cyan, paclitaxel – VDW, ethanol – beads cyan*).**

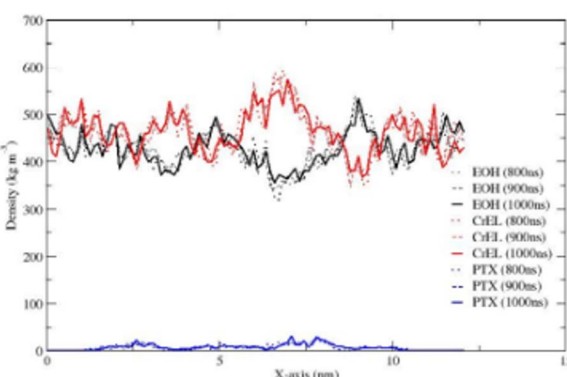

**Fig 5. Partial densities of ethanol, CrEL and paclitaxel in the Taxol micelle along the X-axis integrated over cross-section YZ.**

demonstrates that the conceptual idealization of a "micelle" structure (inner-core-outer corona) as discussed by studies [52,53] is highly unlikely for Taxol.

The partial densities of ethanol, CrEL and paclitaxel along the X-axis are shown in Fig 8 (the Y-, and Z-axis distributions are very similar). The partial densities at times of 800m 900 and 1000ns demonstrate no further leaking of ethanol from the CrEL cluster to surrounding water, and no further water influx into the Taxol micelle. The radial distribution plot in Fig 9 can be used to describe the CrEL-PTX local aggregation process in the mixed solvent system. This indicates that individual PTX molecules are predominately surrounded by CrEL tail groups, even as some ethanol solvent is replaced by water.

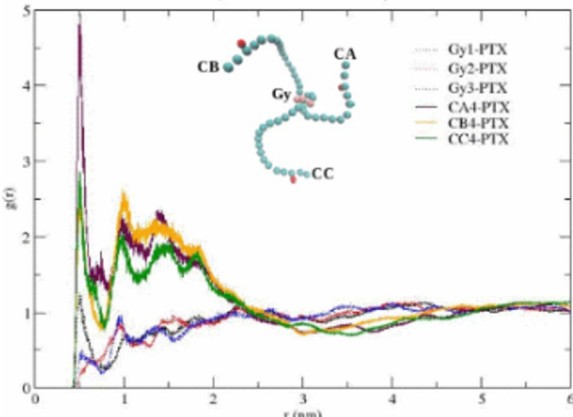

**Fig 6. Radial distribution function for CrEL tails (CA,CB,CC) and CrEL head (Gy) beads with PTX cross-correlation.**

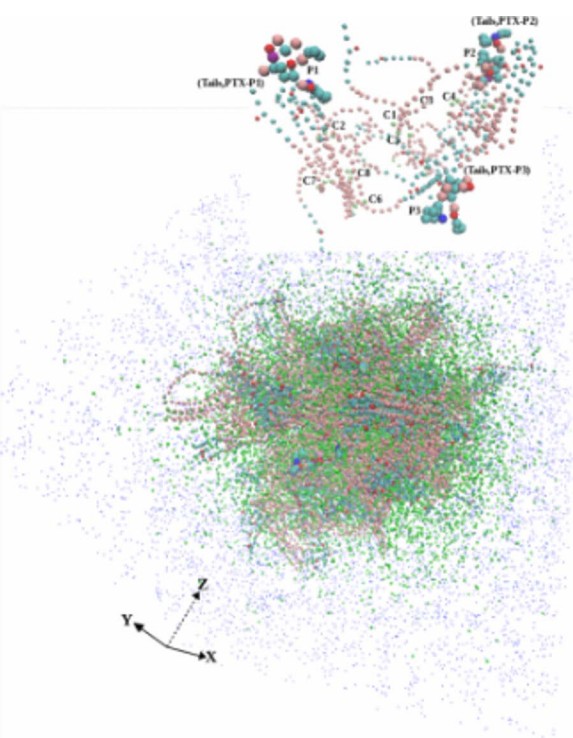

**Fig 7. Taxol micelle in water at 1000ns (an internal spot above showing 3PTX, 10CrEL)** *(CrEL and PTX - VDW, ethanol - beads cyan, water - beads blue, 10% antifreeze water - white)*.

Mean square displacement (MSD) analysis further quantifies Taxol micelle structure and dynamics. Short 20 ns MSD behaviour at different global time periods over the 1000 ns simulation time period can indicate dynamic structure changes. Diffusion constants D(t) were calculated by least squares fitting a straight line (D*t+c) through the MSD(t) over each 20 ns period. For the Taxol micelle in the presence of water, MSD evolution and resulting D(t) for EOH initially change over the first 300 ns before stabilizing by 1000 ns to D of 0.015e-5 cm2/sec. This reflects the early mixing/ exchange of water

and EOH as the Taxol micelle adjusts to the bounding water. In contrast, MSD behaviour and D(t) for the lower mobility PTX and CrEL both fluctuate over time, again indicating local structural changes as PTX molecules aggregate/disaggregate within the micelle over time. This behaviour was observed to be consistent with the spatial distribution and RDF plots shown above.

### Taxol structure with high PTX concentration

In order to get increased conformational statistics on PTX behaviour within a CrEL micelle, we also similarly investigated a high concentration (32 PTX) Taxol micelle scenario. This provides an additional perspective on the stability of PTX within such a micelle. Conformational analysis of both Taxol scenarios is given in section (PTX and CrEL conformations dynamics).

An overall comparison of the simulation results showed that the CrEL aggregation in the high concentration Taxol case was observed to have very similar patterns as that of the standard concentration Taxol micelle case (such as existing CrEL tail clusters in a 3D landscape). VMD visualization showed PTX micro aggregations within these CrEL cavities where initially excess PTX were trapped. In general, we may conclude that the CrEL vehicle can load much higher PTX concentrations than standard concentration with no significant aggregations. (although some dynamic 2–4 microaggregates are seen for the 32PTX case). VMD plots at 1000 ns for the HC micelle cases are very

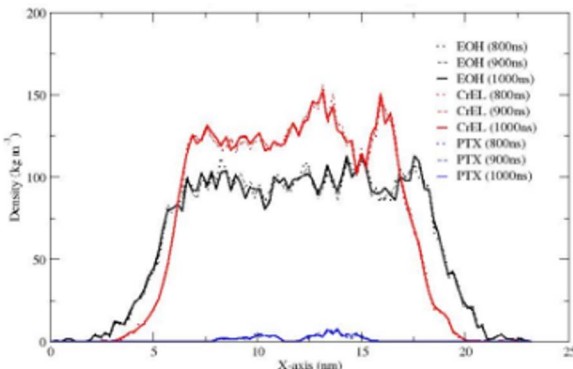

**Fig 8. Partial densities of ethanol, CrEL and PTX in the Taxol micelle with water along the X-axis integrated over cross-section YZ.**

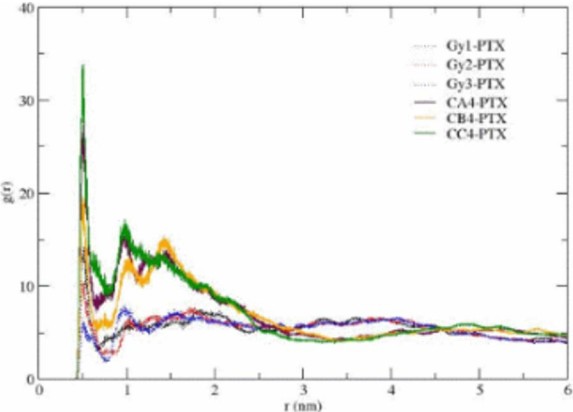

**Fig 9. Radial distribution function for CrEL tails (CA,CB,CC) and CrEL head (Gy) beads with PTX cross-correlation.**

analogous to those shown in Figs 4 and 7 for the SC cases, except for these enhanced microaggregates. S1-S6 Figs in S1 file (as supporting materials) compares MSD behaviours of the SC and HC Taxol micelle. The different aggregation characteristics result in time evolution for the PTX MSD plots, whereby the 8PTX case shows free fluctuations while the 32PTX case stabilizes to smaller MSD behaviour at later times. S7 and S8 Figs in S1 file (as supporting materials) illustrates RDF structure analysis for the HC Taxol micelle and shows a similar structure to the RDFs of the SC micelle described previously.

CG simulations demonstrate that Taxol micelle is not conventional micelle (as formed from linear surfactants or block copolymer aggregation). Our key argument here is that, if there are no other mechanisms (which may be unlikely for AA simulations or laboratory experiments) then CrEL will be aggregated in a nonconventional way in a 3D landscape. The next section shows some more quantitative analysis on CrEL aggregation.

### CrEL aggregation dynamics

The volumes and the radii of CrEL aggregations in Table 1 were estimated in order to fully quantify the CrEL aggregation dynamics. At steady state (t=1000ns), the estimated CrEL volumes in SC-Taxol and HC-Taxol are 856.155 nm$^3$ and 860.240 nm$^3$, respectively. Based on the molar volume (assuming CrEL density of 1.05 g/cm$^3$,), the volume of 200 CrEL molecules is 765.447 nm$^3$. The differences of 90.708 nm$^3$ and 94.793 nm$^3$ are an approximate expansion of 200 CrEL in the SC and HC-Taxol micelles, respectively.

In MD, the CrEL volume in the Taxol micelle (without water) were estimated as follows: equilibrium Taxol (8PTX, 7888EOH, 200CrEL) volume minus equilibrium binary system (8PTX, 7888EOH) volume. The binary system here was set up by removing all 200CrEL from the equilibrium Taxol micelle at 1000ns. MD simulation of this binary system was continued for another 1000ns. One reason for this binary simulation was to further investigate how rapidly PTX molecules aggregate in absence of CrEL, as expected due to the hydrphobic nature of PTX in polar water. The results showed a rapid aggregation (within 200ns) of 8PTX in SC-Taxol into a nearly spherical shape and 32PTX in HC-Taxol into a nearly cylindrical shape. The PTX aggregations shapes were mainly due to the polarity of 24 beads PTX and the number PTX molecules in the systems.

The radius of 200 CrEL molecules aggregation was calculated as, $R_g = \sqrt{\frac{\sum \|r_i^g\|^2 m_i}{\sum m_i}}$, where, $m_i$ is the mass of atom i and $r_i^g$ is the position of atom i with respect to the group center of all 200 CrEL molecules. The group center here was obtained by applying the center of mass of each molecule. The weighted center of mass for each CrEL conformations was defined as: $r_c^j = \frac{\sum m_i r_i}{\sum m_i}$. Subsequently, the group center and the radius of the aggregated CrEL (or radius of gyration) were calculated. For reliability and convenience, we have written a program for our model cases. One can apply GROMACS analysis program "g_gyrate" to measure the radius of gyration. Table 1 summarizes the radius of CrEL aggregations for the Taxol micelle without water (cubical) and with water (nearly spherical) systems. In the next section, we calculate the radius of conformations for each PTX and CrEL molecules.

Experimentally the size of a Taxol micelle has been shown to be 13 nm diameter (6.5 nm radius) [54]. This implies an experimental volume of the micelle (= $4/3\pi R_{total}^3$) of 1150 nm$^3$. The calculated micelle results (radius, volume) are therefore

**Table 1. Volume and radius of CrEL aggregation.**

| Simulation | SC-Taxol | | SC-Taxol in Water | HC-Taxol | | HC-Taxol in Water |
|---|---|---|---|---|---|---|
| t (ns) | V (nm$^3$) | Rg (nm) | Rg (nm) | V (nm$^3$) | Rg (nm) | Rg (nm) |
| 800 model | 856.235 | 6.023 | 5.753 | 860.644 | 6.030 | 5.810 |
| 900 | 855.855 | 6.017 | 5.745 | 860,610 | 6.035 | 5.800 |
| 1000 | 856.155 | 6.023 | 5.735 | 860.240 | 6.039 | 5.785 |

roughly consistent with the experimental Taxol micelle values. However, the MD analysis presented in this paper indicates this naïve picture is very far from the correct structure for the Taxol delivery structure. Our MD simulations indicate this micelle structural difference can be traced to the unique properties of the 3-winged hydrophilic nature of the CrEL surfactant.

In order to quantify the evolution of the Taxol micelle, we have analyzed the root mean square deviation (RMSD) and the radius of gyration (Rg), progressively, at every 20ns over the entire simulation period of 1000ns. Table 1 showed no noticeable changes in Rg values when other variables (such as mean square displacement, radial distribution function) also described the Taxol micelle equilibrium state. To obtain a concrete evolution picture, we have analyzed Pearson and Spearman's coefficients for RMSD for CrEL and PTX. These correlation coefficients showed a significant large positive relationship and nearly stabilized around simulation time of 1000ns with Pearson correlation of $r_p = 0.973$ (Pearson) for CrEL and $r_p = 0.834$ (Pearson) for PTX. Analogously, the Spearman rank correlation coefficients of $r_s = 0.970$ (Spearman) for CrEL and $r_s = 0.770$ (Spearman) for PTX were found. The S3 Table in S1 file (as supplementary materials) presents the complete evolution of Pearson and Spearman correlation coefficients. This study shows the utility of these two variables (RMSD, Rg) to quantify Taxol micelle dynamics.

## PTX and CrEL conformations dynamics

Numerous NMR and electron crystallographic studies for PTX in solution coupled to conformational analysis have led to a range of suggestions for both the binding mode and the bioactive conformation of PTX including a T-Taxol (butterfly) conformer [31]. Conversely, to the best of our knowledge, there is no such conformational analysis for CrEL.

In our work, one of the main objectives is to simulate reliable conformational properties for PTX and CrEL in solution using both AA and CG methods. Here, we have isolated all the PTX and CrEL conformations from the MD cases. After isolating all the conformations, the global minimum potential energies for each molecular conformation were predicted, separately, by conducting energy minimization simulations. As potential energy is gauge dependent, the energy minimization simulations were performed in the same box by center positioning each PTX and CrEL. And the relative energy for each molecule was then defined by subtracting the lowest conformation energy. This setting showed no changes in the MD simulated conformations structures during the energy minimization. The energy minimization was accomplished using a steepest descent algorithm, with convergence achieved once the maximum force on any one atom was less than 100 kJ. The Verlet cut-off scheme was employed using the default Verlet buffer size set to 0.005 kJ.mol$^{-1}$.ps$^{-1}$ energy drift per atom. It is assumed that the total energy drift occurring in the system is usually much smaller than this default setting for the estimate.

The above analysis includes: four binary cases (8 PTX in ethanol, 8 PTX in water, 50 CrEL in ethanol and 50 CrEL in water) and one ternary Taxol micelle (32PTX-7888EOH-200CrEL). By conducting systematic AA and CG simulations, MD equilibrium states for all cases were obtained. For each equilibrium system, PTX and CrEL molecular conformations were then collected. The conformation dynamics of 32 PTX and 200 CrEL molecules in the ternary Taxol micelle are presented step by step in the next section (PTX and CrEL Conformations Dynamics). The relative conformational energies are very good indicators of the dynamics of the MD systems discussed in the previous section.

### AA Taxol micelle - energy conformational status

MD simulation for the AA Taxol micelle was conducted to 50ns (which is a much earlier state in the equilibrium process) extending the premixing simulations of the individual phase creation cubes (section taxol micelle simulation setup). Fig 10 shows the atomic coordinates snapshot of the end of 50ns simulation. Although clearly not at equilibrium, this semi-mixed state is sufficient for a statistical analysis of molecularly relaxed PTX and CrEL chemicals individually. Indeed, monitored

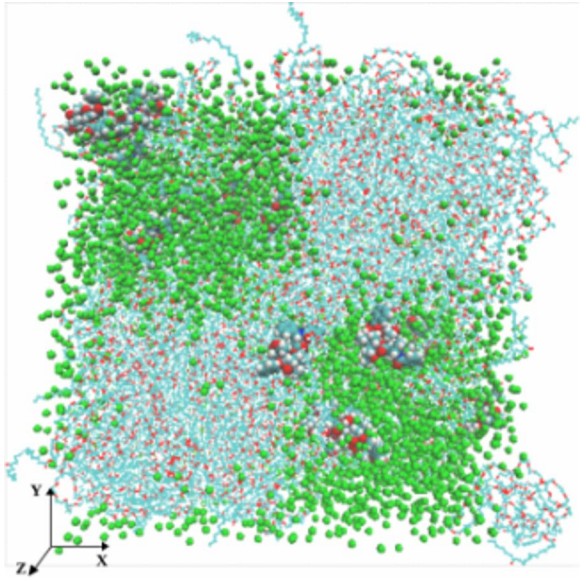

**Fig 10. Spatial distributions of the components of AA Taxol micelle with 32 PTX** *(PTX - VMD, EOH - green bead, CrEL - line (C - cyan, O - red, H - white)).*

molecular conformation stabilized sufficiently for our subsequent conformational analysis. Here, we have calculated all the bonded and non-bonded energy terms at this stage of this AA ternary Taxol micelle. The statistical variation of these components is summarized in Table 2. The detailed statistical analysis for the molecular conformations and conformation energies are presented next. CG Taxol micelle at 1000ns discussed in the earlier section was used for CG molecular conformations study, even though the CG conformation stabilized after 100ns.

**Table 2. The Relative Energies of the PTX and CrEL Conformations.**

| Energy Components | PTX Conformations Energies | | CrEL Conformations Energies | |
|---|---|---|---|---|
| | *(kJ.mol-1)* | | *(kJ.mol-1)* | |
| | **Mean** | **Standard Deviation** | **Mean** | **Standard Deviation** |
| ΔCoulomb-SR | 17.497 | 9.743 | 190.862 | 56.666 |
| ΔCoulomb-14 | 9.206 | 4.647 | 53.947 | 25.439 |
| ΔCoulomb-LR | 3.626 | 1.786 | 59.325 | 27.492 |
| ΔLJ-SR | 18.195 | 11.216 | 230.955 | 85.722 |
| ΔLJ-14 | 15.040 | 7.180 | 19.387 | 9.248 |
| Δbond | 5.263 | 3.266 | 3.724 | 1.658 |
| Δangle | 31.679 | 16.410 | 52.406 | 24.153 |
| Δdihedral | 18.411 | 10.938 | 47.754 | 22.663 |
| ΔCoulomb-Sum | 12.372 | 7.592 | 219.547 | 68.492 |
| ΔLJ-Sum | 32.641 | 13.678 | 233.748 | 83.969 |
| Δnon-bonded | 23.098 | 12.795 | 454.412 | 148.160 |
| Δbonded | 37.466 | 17.708 | 67.684 | 37.688 |

## PTX conformations (AA vs CG)

PTX molecule is a complex diterpenoid with a conformationally immobile core consisting of the fused rings and flexible side chains. Lakdawala et al. [31] calculated AA relative energies using energy minimization simulations for several Taxol conformations derived from NMR and X-ray analyses with a set of widely used force fields. In this study, we have analyzed the PTX conformations in ethanol, water and Taxol micelle and subsequently calculated the conformations energies. Here, we employed the force field OPLSA for AA MD and MARTINI for CG MD simulations again utilizing energy minimization techniques. While Lakdawala et al. [31] analyzed 7 selected PTX conformations in gas phase or with continuum solvent models, our conformational analyses are conducted in more realistic dynamic explicit multi-solvent Taxol scenarios with a much larger 32 conformational set, bracketing the 7 conformations of Lakdawala et al. [31].

In the ternary Taxol micelle (32PTX-7888EOH-200CrEL), we have analyzed 32PTX conformations energies. Table 3 summarizes the relative conformation energies of 32 PTX molecules in the Taxol micelle. The conformations of AA PTX #28 and CG PTX #22 showed the lowest potential energies of 472.144 kJ.mol$^{-1}$ and 9.871 kJ.mol$^{-1}$, respectively. These energies were then taken as the reference energies to define relative energies (U-U$^{ref}$) for all PTX in AA and CG systems. The average relative energies are 19.199 kJ.mol$^{-1}$ for AA and 34.043 kJ.mol$^{-1}$ for CG. Fig 11 showed the molecular conformations for 2 AA and 2 CG PTX molecules, respectively. The average conformations here showed the distances between the centroids of the side chain phenyl rings (1, 3) and the terminus ring (2). Table 3 also reports the radii of conformation for 32 PTX molecules individually from AA and CG simulations.

Using all conformations energy data, AA and CG PTX energy histograms are plotted in Fig 12. This plot illustrates that the relative energy varies from 0.0 to 70 kJ.mol$^{-1}$. In the AA case, 12 PTX (higher frequency) were at energy between 30–40 kJ.mol$^{-1}$, 5 PTX at lower energy (10–20 kJ.mol$^{-1}$), and only one at higher energy (50–60 kJ.mol$^{-1}$). In CG, 10 PTX (higher frequency) were found at energy between 20–30 kJ.mol$^{-1}$, one PTX at lower energy (0–10 kJ.mol$^{-1}$), and one at higher energy (60–70 kJ.mol$^{-1}$). These histograms can be described as slightly right skewed Gaussian distributions. The mean, median and standard deviation *(in kJ.mol$^{-1}$)* are μ=19.199, M=18.845, σ=9.556 for AA and μ=34.043, M=31.469, σ=14.915 for CG. The results show a nearly similar PTX relative energy distribution in AA and CG. It may be important to note that CG conformation energies were reduced from AA to lower energy states (~450 kJ.mol$^{-1}$), attributable to the change in force fields (AA to CG). This energy drop is not shown in the histogram plots due to subtracting the reference energy (as U-U$^{ref}$). Lakdawala et al. [31] reported that when several groups in PTX converted to their hydrocarbon analogs (i.e., also an effective force field change), the optimized structures were unchanged but the individual global minima were shifted for different conformers. Our detailed statistical analysis of AA and CG energy conformational data support this concept.

## CrEL conformations (AA vs CG)

In this study, we have quantified the conformations and the energy states for 200 CrEL molecules in the AA and Taxol micelles. We recognize that CrEL may have many conformation properties depending on its interaction with the excipients (such as water and ethanol). The CrEL molecule has three wings (head of each wing is 12 monomers of PEG and tail is ricinoleic acid). PEG is highly flexible and has a tendency to fold due to its repeating monomers. Conversely, ricinoleic acid is not very flexible compared with PEG. Because of their composite three-winged molecular structure, CrEL molecules may exhibit many conformation properties. The challenge is how to quantify such vital properties for a given application. The three wings of the CrEL molecule have very complex conformation properties. The central polar segments have twisted conformations and the nonpolar tail segments have strong clustering properties.

From the simulated Taxol micelle, 200 CrEL molecules were first separated and then potential energy for each was obtained by conducting energy minimization simulations. We were able to develop a correlation between potential energy and conformation (how CrEL potential energy changes with CrEL conformation). The spiral shape conformation was observed to be the dominant species. Fig 13 shows the selected spiral conformations for AA and CG CrEL,

**Table 3. AA and CG PTX conformations and the relative changes in the conformational energy.**

| PTX | AA MD | | | | CG MD | | | |
|---|---|---|---|---|---|---|---|---|
| | U-Uref (kJ.mol-1) | L12 (A) | L13 (A) | RC(A) | U-Uref (kJ.mol-1) | L12 (A) | L13 (A) | RC(A) |
| 1 | 16.906 | 11.375 | 8.216 | 5.118 | 13.010 | 10.262 | 7.070 | 7.187 |
| 2 | 17.540 | 12.479 | 7.513 | 5.293 | 50.376 | 12.626 | 5.792 | 7.625 |
| 3 | 24.857 | 10.948 | 6.799 | 4.932 | 32.455 | 7.328 | 9.348 | 6.928 |
| 4 | 2.420 | 11.333 | 6.962 | 5.016 | 20.122 | 16.051 | 10.892 | 7.186 |
| 5 | 21.343 | 11.394 | 7.421 | 5.059 | 26.053 | 7.328 | 11.166 | 6.140 |
| 6 | 7.462 | 12.032 | 7.112 | 5.143 | 39.334 | 13.325 | 11.619 | 6.673 |
| 7 | 19.308 | 11.054 | 7.605 | 5.154 | 23.500 | 10.665 | 8.766 | 6.348 |
| 8 | 40.921 | 11.673 | 6.902 | 5.049 | 52.827 | 10.124 | 13.865 | 7.099 |
| 9 | 27.304 | 11.120 | 8.268 | 5.106 | 27.053 | 5.647 | 10.917 | 7.146 |
| 10 | 24.003 | 11.770 | 6.645 | 5.144 | 43.504 | 10.275 | 11.130 | 6.832 |
| 11 | 29.486 | 12.414 | 7.621 | 5.246 | 32.193 | 13.925 | 6.187 | 7.178 |
| 12 | 4.250 | 11.295 | 9.762 | 4.868 | 21.352 | 5.668 | 9.963 | 6.689 |
| 13 | 19.581 | 10.240 | 7.828 | 5.064 | 28.981 | 12.367 | 4.981 | 6.733 |
| 14 | 5.286 | 11.019 | 7.356 | 4.999 | 12.569 | 11.894 | 5.108 | 6.975 |
| 15 | 16.244 | 11.647 | 7.907 | 5.288 | 28.302 | 8.510 | 5.415 | 6.507 |
| 16 | 16.961 | 11.852 | 7.692 | 5.225 | 38.986 | 13.156 | 8.872 | 6.345 |
| 17 | 10.159 | 11.663 | 8.379 | 5.202 | 52.947 | 15.409 | 9.478 | 7.536 |
| 18 | 27.010 | 11.431 | 6.197 | 5.244 | 53.010 | 17.132 | 10.507 | 7.950 |
| 19 | 27.429 | 11.817 | 9.267 | 5.054 | 27.653 | 18.571 | 13.963 | 7.746 |
| 20 | 22.796 | 10.551 | 7.811 | 4.979 | 40.243 | 16.687 | 11.915 | 6.381 |
| 21 | 10.766 | 12.252 | 7.413 | 5.135 | 45.035 | 15.505 | 15.798 | 7.855 |
| 22 | 26.638 | 10.807 | 7.046 | 4.930 | 0.000 | 4.941 | 10.005 | 6.318 |
| 23 | 31.951 | 10.608 | 7.349 | 4.989 | 25.069 | 7.248 | 9.774 | 6.581 |
| 24 | 30.498 | 10.565 | 6.400 | 4.925 | 42.266 | 11.491 | 11.005 | 6.838 |
| 25 | 10.081 | 10.742 | 8.497 | 5.026 | 28.721 | 5.741 | 9.360 | 6.466 |
| 26 | 13.624 | 10.984 | 8.281 | 5.097 | 26.662 | 12.157 | 6.017 | 7.216 |
| 27 | 16.894 | 12.212 | 7.493 | 5.174 | 46.781 | 17.922 | 14.109 | 7.627 |
| 28 | 0.000 | 11.726 | 7.243 | 5.031 | 30.746 | 12.382 | 6.171 | 6.823 |
| 29 | 24.587 | 11.549 | 7.303 | 4.990 | 47.000 | 15.952 | 11.372 | 6.752 |
| 30 | 16.506 | 11.323 | 6.605 | 4.920 | 13.455 | 8.792 | 6.044 | 6.797 |
| 31 | 33.185 | 10.795 | 9.078 | 5.161 | 74.304 | 18.106 | 15.267 | 7.796 |
| 32 | 18.381 | 12.730 | 7.862 | 5.084 | 44.859 | 8.954 | 14.101 | 7.673 |
| Ave | 19.199 | 11.419 | 7.620 | 5.083 | 34.043 | 11.754 | 9.874 | 6.998 |

$U^{ref}$=472.144 kJ.mol$^{-1}$ (AA), 9.871 kJ.mol$^{-1}$ (CG)

respectively. The molecule reached a high energy state when wings were fully spread, and at low energy state when wings were fully closed.

Using 200 CrEL energy data, frequency histogram was plotted by dividing the entire energy spectrum into several energy groups. Fig 14 shows the relative energy histograms of the 200 AA and CG CrEL conformations data. The frequency histograms here can be described as left skewed Gaussian distributions. The mean, median and

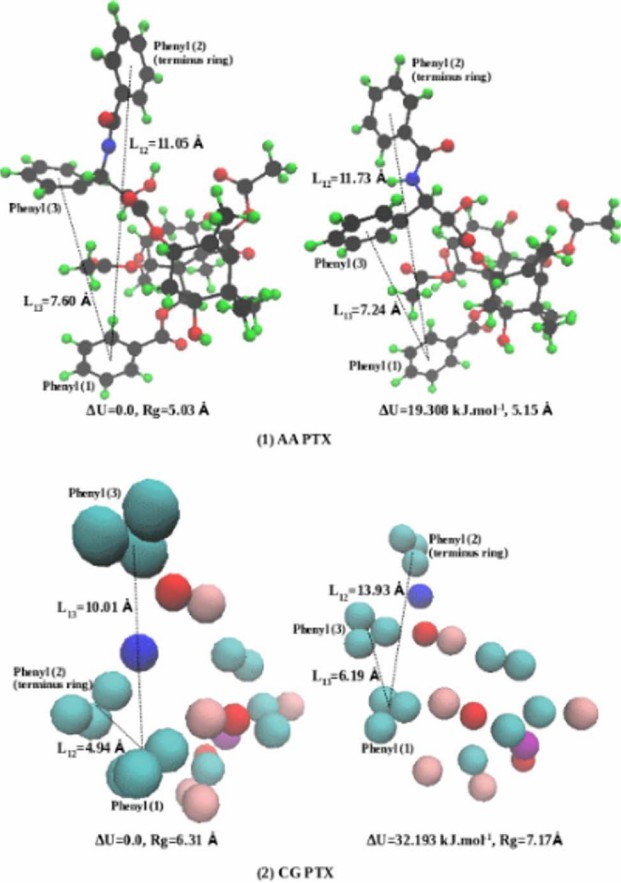

**Fig 11. AA and CG PTX molecular conformations showing the distances between the centroids of the side chain phenyl rings (1, 3) and the terminus ring (2).**

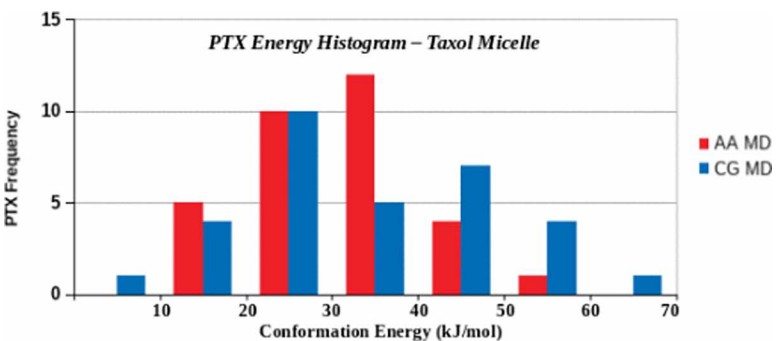

**Fig 12. Distribution of 32 PTX conformations energies in the Taxol micelle (32PTX, 7888EOH, 200CrEL).**

standard deviation *(in kJ.mol⁻¹)* are μ=342.008, M=344.025, σ=127.928 for AA MD and μ=352.482, M=370.750, σ=113.358 for CG MD.

This histogram describes that the relative energy of CrEL molecules varies from 0.0 to 650 kJ.mol⁻¹. In AA case, 59 CrEL (higher frequency, mostly wings formed spiral shapes) were at energy between 350–450 kJ.mol⁻¹, 3 CrEL at lower

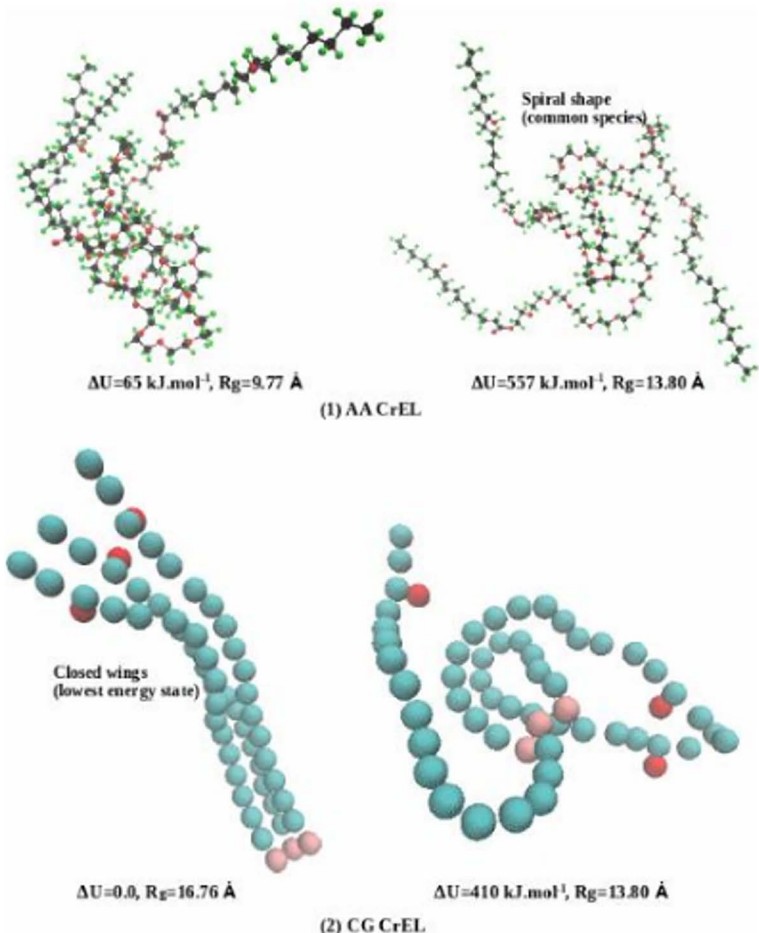

**Fig 13. AA and CG CrEL conformations in the Taxol micelle (CrEL at the highest energy state when wings are fully spread, not shown, and lowest energy state when wings are fully closed, dominated spiral shapes).**

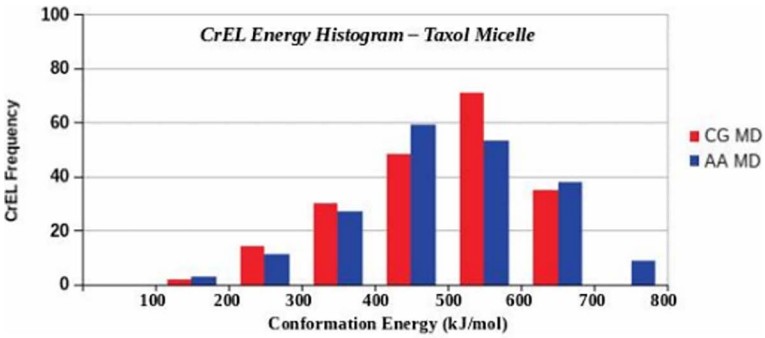

**Fig 14. Distribution of 200 CrEL molecular conformations energies in the 32PTX Taxol micelle.**

energy (50–150 kJ.mol$^{-1}$) (wings were closed), and 9 at higher energy (650–760 kJ.mol$^{-1}$) (wings were fully spread). In CG, 71 CrEL (higher frequency, most wings formed spiral shapes) were found at energy between 450–550 kJ.mol$^{-1}$, 2 CrEL at lower energy (50–150 kJ.mol$^{-1}$) (wings were fully closed), and 35 at higher energy (550–6500 kJ.mol$^{-1}$) (wings were spread). The results show a nearly similar CrEL relative energy distribution in AA and CG.

CrEL conformational information from AA and CG simulations analogous to the PTX information in Table 3 has also been analyzed by us but not reported here due to large volume of data (200 CrEL molecules). The statistical distribution of the 200 CrEL conformational radii was observed to be consistent with the energy distributions in Fig 14.

In order to quantify how CrEL molecules interact with solvent ethanol and water, the relative conformation energies of the 50 CrEL molecules in two binary cases were analyzed in S2 file (as supporting materials). It should be noted that, in comparison with AA, the CG conformation energies were shifted significantly to lower energy states. Furthermore, we have analyzed our initial simulated CrEL phase containing 50 AA CrEL molecules. The results showed a narrow Gaussian distribution with mean conformation energy at 400 kJ.mol$^{-1}$.

## Drug delivery conformation summary

The average conformational energies of CrEL are seen to be multiples of PTX conformational energies. This can be attributed to the increased number of heavy atoms in CrEL versus PTX (roughly three times) and the resultant increased factorial interaction between atoms. The lowest conformation of PTX corresponds to the centre rigid portion of PTX surrounded by closed phenyl rings (i.e., a butterfly with closed wings). The more probable PTX conformation (with higher energy) has these wings more separated and somewhat twisted (i.e., a butterfly with open wings). This is confirmed by our spatial plots, Fig 11 and S4-S5 Tables in S2 file (as supporting materials). Correspondingly, the lowest conformational energies for CrEL imply linear alignment of its three wings, while the more probable conformation corresponds to the spreading of these wings, into an almost helical pattern (as confirmed by our spatial plots, Fig 13 and S10-S13 Figs in S2 file (as supporting materials). These provide useful qualitative summaries of our quantitative conformational analysis.

More recently, research and clinical trials have focused on alternate nanoparticle formulations other than Taxol to deliver paclitaxel. Each comes with their own drug delivery characteristics and issues. All studies, or at least the grand majority, compare their delivery characteristics with those of Taxol. A good review of such studies has been presented by Sofias et al. [55]. Here they emphasize the growing acceptance of Abraxane (a 130 nm albumin-based nanoparticle) as a viable Taxol alternative. An example study comparing such behaviours has been conducted by Chen et al. [56]. Other nanoparticle formulations for paclitaxel (including liposomes) are referenced by Sofias et al. [55]. Finally, these comparisons have recently been further extended to carriers for drugs other than paclitaxel, as reviewed by Spada et al. [57], again using Taxol as a basis for comparison. Our methods should be applicable to these systems as well.

## Conclusions and recommendations

Paclitaxel (PTX) is a highly successful anti-neoplastic cancer drug, most often using the first clinically successful PTX-delivery method with Cremophor EL (CrEL) and ethanol (EOH), here termed the "Taxol micelle". PTX molecule is a complex diterpenoid with a conformationally immobile core consisting of the fused rings and flexible side chains. The CrEL molecule has three wings (head of each wing is 12 monomers of PEG and tail is ricinoleic acid). PEG is highly flexible and has a tendency to fold due to its repeating monomers. Conversely, ricinoleic acid is not very flexible compared with PEG. As a result, CrEL may have many conformation properties depending on its interaction with the excipients (such as water and ethanol). We developed our models in a stepwise fashion, considering first binary mixtures of both chemicals PTX and CrEL with the solvents ethanol (EOH) and water, comparing AA and CG representations. Thereafter the ternary Taxol micelle (PTX-CrEL-EOH) was investigated without or with surrounding water, again comparing the consistency of structures at short time (50 ns) using both AA and CG methods. Thereafter long time (1000 ns) CG simulations allow mixing and aggregation to be further quantified. The long time CG simulations demonstrated PTX and CrEL aggregation

dynamics in the micelle. Our CG simulations can furthermore be reversed-mapped into AA equivalents, using an algorithm discussed in detail in a future publication. The results showed a highly varied CrEL aggregation landscape where non-polar CrEL tails clustered inside and the boundaries of the micelle. PTX molecules were locally trapped into the CrEL tail landscape and remained essentially dispersed (no aggregation). Our step-by-step investigations include: PTX dimer formation as well as mixing of ethanol and water over time. One of the important observations was to be an oscillatory behaviour of the dispersed PTX in the steady state Taxol micelle. The high concentration Taxol simulations showed that PTX mostly dispersed within CrEL clusters. CrEL formed an aggregation network in 3D landscape. Detailed analysis showed that PTX molecules are dynamically trapped within the cavities of CrEL (i.e., repeated patterns of attached/released with CrEL particles), Comparing with the standard concentration (8PTX) Taxol micelle results, CrEL can load much higher PTX than standard concentration without any observable PTX aggregation within the timescale of our CG simulation. Taxol micelles have its unexpected non-standard micelle structure due to CrEL specific properties.

To fully characterize the CrEL and ethanol mixing and paclitaxel solubility, we have estimated that the relative potential energy limit of a single CrEL molecule is about 600 kJ.mol$^{-1}$. This energy in average is about 400 kJ.mol$^{-1}$ for the synthesized pure CrEL phase. The relative potential energy of a single PTX molecule is about 60 kJ.mol$^{-1}$. By applying an energy minimization algorithm, we obtained the global minimum conformations energies for each PTX and CrEL molecules for four binary cases (8 PTX in ethanol, 8 PTX in water, 50 CrEL in ethanol and 50 CrEL in water) and ternary Taxol micelle (32PTX, 7888EOH, 200CrEL). The average physical conformations for all PTX molecules were described by the distances between the centroids of the phenyl rings (2, 3) and the terminus phenyl ring (1). Three wing CrEL molecules on the other hand have very complex physical conformations properties. The central polar segments have twisted conformations and the nonpolar tail segments have strong clustering properties. For two binary cases, we calculated six variables to define the physical shapes of CrEL molecules. In general, wings were fully spread at highest relative conformation energy, wings were fully closed at lowest energy and dominant shapes were observed to be spiral. We have summarized all PTX and CrEL conformations using energy frequency histograms and provided full statistical analysis.

While numerous research and clinical articles heuristically refer to the "Taxol micelle" in the past, ours is the first to investigate its actual structure – it is not a conventional micelle as we have shown. We have furthermore analyzed in detail the conformational behaviour of its drug components PTX and CrEL within the micelle, extending limited studies of PTX conformations and non-existing studies of CrEL conformations. These behaviours fundamentally contribute to the stability of Taxol and its ability to propagate throughout the body. Our simulation methodology and the dynamics sensitivity analysis presented in this paper can also be used to analyze and understand the microscopic effectiveness of these other drug formulations. Most particularly, the analysis presented here for Taxol modelling is directly applicable to the Taxotere micelle, with docetaxel as an alternative to paclitaxel and polysorbate 80 replacing CrEL as carrier molecule [58], resulting in observable differences in drug stability and drug effectiveness.

## Supporting information

**S1 File. Taxol micelle.** Taxol micelle analysis (S1_File.pdf): more detailed analysis of the time evolution of PTX and CrEL interactions in the Taxol micelle is presented. This includes mean square displacements for PTX and CrEL, plus radial distribution functions illustrating the interactions of PTX with the head and tail groups of the CrEL molecules.
(PDF)

**S2 File. Molecular conformation.** Molecular conformation analysis (S2_File.pdf): details of PTX and CrEL molecular conformations are analyzed for various binary PTX and CrEL systems, as well as ternary Taxol micelle cases, comparing both AA and CG molecular conformations.
(PDF)

**S3 File. PTX data.** PTX data(S3_File.pdf): CG and AA data for PTX molecules were included as supporting materials. (PDF)

**S4 File. CrEL data.** CrEL data (S4_File.pdf): CG and AA data for CrEL molecules were included as supporting materials. (PDF)

AcknowledgmentWe thank the reviewers for their critical review and very thoughtful comments. The authors appreciate the editor for handling the paper and getting it reviewed.

## Author contributions

**Conceptualization:** Mafiz Uddin, Dennis Coombe.

**Data curation:** Mafiz Uddin, Dennis Coombe.

**Formal analysis:** Mafiz Uddin, Dennis Coombe.

**Funding acquisition:** Dennis Coombe.

**Investigation:** Mafiz Uddin, Dennis Coombe.

**Methodology:** Mafiz Uddin, Dennis Coombe.

**Project administration:** Mafiz Uddin.

**Resources:** Mafiz Uddin, Dennis Coombe.

**Software:** Mafiz Uddin, Dennis Coombe.

**Supervision:** Mafiz Uddin.

**Validation:** Mafiz Uddin, Dennis Coombe.

**Visualization:** Mafiz Uddin, Dennis Coombe.

**Writing – original draft:** Mafiz Uddin, Dennis Coombe.

**Writing – review & editing:** Mafiz Uddin, Dennis Coombe.

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
