## [Decision Letter · Decision Letter 0]

19 Jan 2025

Dear Dr. Uddin,

Thank you for submitting your manuscript to PLOS ONE. After careful consideration, we feel that it has merit but does not fully meet PLOS ONE’s publication criteria as it currently stands. Therefore, we invite you to submit a revised version of the manuscript that addresses the points raised during the review process.

We look forward to receiving your revised manuscript.

Kind regards,

Abdullahi Ibrahim Uba

Academic Editor

PLOS ONE

Journal Requirements:

**Additional Editor Comments:**

The discussion section can be improved by incorporating recent studies characterizing the dynamic behavior of Paclitaxel and Cremophor EL

Reviewers' comments:

Reviewer's Responses to Questions

**Comments to the Author**

1. Is the manuscript technically sound, and do the data support the conclusions?

Reviewer #1: Yes

2. Has the statistical analysis been performed appropriately and rigorously?

Reviewer #1: Yes

3. Have the authors made all data underlying the findings in their manuscript fully available?

Reviewer #1: Yes

4. Is the manuscript presented in an intelligible fashion and written in standard English?

Reviewer #1: Yes

Reviewer #1: The manuscript has sufficient details to be considered for publication in PLOS One, provided that the following comment is properly addressed:

The RMSD (Root Mean Square Deviation) and Radius of Gyration plots versus time must be analyzed and discussed in depth.

**Do you want your identity to be public for this peer review?** For information about this choice, including consent withdrawal, please see our Privacy Policy

Reviewer #1: No

---

## [Author Response · Author response to Decision Letter 0]

30 Jan 2025

We have addressed the reviewer’s and editor’s comments point by point and revised the manuscript accordingly (see our attachments "Response_to_Revewers.pdf").

The authors wish to again thank the editor and reviewer for very thoughtful comments. We have made every effort to respond to the issues they have raised.

---

## [Decision Letter · Decision Letter 1]

2 Feb 2025

Drug Delivery Process Simulation - Quantifying the Conformation Dynamics of Paclitaxel and Cremophor EL

PONE-D-24-49703R1

Dear Dr. Mafiz Uddin,

We’re pleased to inform you that your manuscript has been judged scientifically suitable for publication and will be formally accepted for publication once it meets all outstanding technical requirements.

Kind regards,

Abdullahi Ibrahim Uba, PhD

Academic Editor

PLOS ONE

Additional Editor Comments (optional):

Reviewers' comments:

Reviewer's Responses to Questions

**Comments to the Author**

Reviewer #1: (No Response)

2. Is the manuscript technically sound, and do the data support the conclusions?

Reviewer #1: (No Response)

3. Has the statistical analysis been performed appropriately and rigorously?

Reviewer #1: (No Response)

4. Have the authors made all data underlying the findings in their manuscript fully available?

Reviewer #1: (No Response)

5. Is the manuscript presented in an intelligible fashion and written in standard English?

Reviewer #1: (No Response)

Reviewer #1: (No Response)

**Do you want your identity to be public for this peer review?** For information about this choice, including consent withdrawal, please see our Privacy Policy

Reviewer #1: No

---

## [Editor Report · Acceptance letter]

PONE-D-24-49703R1

PLOS ONE

Dear Dr. Uddin,

I'm pleased to inform you that your manuscript has been deemed suitable for publication in PLOS ONE. Congratulations! Your manuscript is now being handed over to our production team.

Kind regards,

on behalf of

Dr. Abdullahi Ibrahim Uba

Academic Editor

PLOS ONE